# Cross-Country Differences in Stay-at-Home Behaviors during Peaks in the COVID-19 Pandemic in China and the United States: The Roles of Health Beliefs and Behavioral Intention

**DOI:** 10.3390/ijerph18042104

**Published:** 2021-02-21

**Authors:** Wei Hong, Ru-De Liu, Yi Ding, Jacqueline Hwang, Jia Wang, Yi Yang

**Affiliations:** 1Beijing Key Laboratory of Applied Experimental Psychology, National Demonstration Center for Experimental Psychology Education (Beijing Normal University), Faculty of Psychology, Beijing Normal University, Beijing 100875, China; psyhongwei@163.com (W.H.); 201921061092@mail.bnu.edu.cn (Y.Y.); 2Graduate School of Education, Fordham University, New York, NY 10023, USA; yding4@fordham.edu (Y.D.); jhwang26@fordham.edu (J.H.); 3Teachers’ College, Beijing Union University, Beijing 100874, China; wangjia@mail.bnu.edu.cn

**Keywords:** cross-country difference, stay-at-home behavior, health belief, behavioral intention, college student

## Abstract

The novel coronavirus disease 2019 (COVID-19) rapidly escalated to a global pandemic. To control the rate of transmission, governments advocated that the public practice social distancing, which included staying at home. However, compliance with stay-at-home orders has varied between countries such as China and the United States, and little is known about the mechanisms underlying the national differences. Based on the health belief model, the theory of reasoned action, and the technology acceptance model, health beliefs and behavioral intention are suggested as possible explanations. A total of 498 Chinese and 292 American college students were recruited to complete an online survey. The structural equation modeling results showed that health beliefs (i.e., perceived susceptibility, severity, and barriers) and behavioral intention played multiple mediating roles in the association between nationality and actual stay-at-home behaviors. Notably, the effect via perceived barriers → behavioral intention was stronger than the effects via perceived susceptibility and severity → behavioral intention. That is, American participants perceived high levels of susceptibility whereas Chinese participants perceived high levels of severity, especially few barriers, which further led to increased behavioral intention and more frequent stay-at-home behaviors. These findings not only facilitate a comprehensive understanding of cross-country differences in compliance with stay-at-home orders during peaks in the COVID-19 pandemic but also lend support for mitigation of the current global crisis and future disease prevention and health promotion efforts.

## 1. Introduction

The novel coronavirus disease 2019 (COVID-19) rapidly spread throughout China and subsequently escalated into a global pandemic. According to the World Health Organization (WHO) [1], the transmission modes of this disease mainly include contact, droplet, and airborne transmission. To inhibit the widespread transmission of the coronavirus, public health experts and government officials have advocated practices that include handwashing hygiene, wearing facial masks, maintaining physical distancing, and restricting unnecessary movement [2,3]. Within these measures, staying at home has been recommended as an effective way to avoid contracting and spreading the disease [4,5] and has been implemented as government policy in China and the United States, among other countries [6,7]. However, varied levels of compliance with lockdown policies during peaks in COVID-19 have manifested national differences associated with cultural orientation. People in some countries have consistently complied with stay-at-home orders, but people in other countries have done so inconsistently. The plausible explanations underlying the cross-country differences remain unknown and have stimulated pertinent research.

Based on the extant literature, researchers have proposed cognitive processes such as health beliefs to mediate the associations between individual characteristics (e.g., ethnicity, cultural orientation, and traits) and health behaviors according to the health belief models [8,9]. Similarly, behavioral intention has been considered as a salient mediator to link cognitive beliefs and actual behaviors as described in the theory of reasoned action [10] and the technology acceptance model [11]. Therefore, nationality may be associated with engagement in stay-at-home behaviors through the effects of health beliefs and behavioral intention in the context of the COVID-19 pandemic. However, few studies have attempted to examine these relations from an empirical perspective. To this end, this study aimed to investigate the cross-country differences between China and the United States in disease-preventive behaviors and its potential mechanisms from the perspective of cognitive beliefs and behavioral intention, which lend support for efforts to mitigate the global COVID-19 pandemic.

### 1.1. Stay-at-Home Behaviors in China and the United States

As a type of demographic characteristic, nationality was argued to be associated with engagement in preventive behaviors in the context of the 2003 severe acute respiratory syndrome (SARS) epidemic [12]. In particular, Hofstede et al. [13] noted that different nations are rooted in different cultural values; for example, China is known as a typical collectivist country while the United States is known as a typical individualistic country. According to the pathogen prevalence hypothesis, collectivism (in contrast to individualism) functions as a protective factor against pathogen transmission partially because collectivism emphasizes tradition and conformity while individualism emphasizes personal autonomy and independence [14,15]. That is, compared with individualists, collectivists are more likely to comply with authorities and less likely to deviate from social norms [16], which facilitates the group’s adaptive disease-preventive behaviors in the face of an epidemic. In a 2020 study, Huang, Ding, Liu, Wu, Zhu, Li, and Zhu [15] found that collectivism positively predicted preventive behaviors such as staying at home during the COVID-19 outbreak. Furthermore, collectivists are characterized by high conscientiousness [17], which is associated with frequent pandemic-related rule-respecting behaviors, including staying at home [18]. Based on these findings, we propose the following hypothesis (see Figure 1):

**Hypothesis** **1** **(H1).**
*Nationality is associated with engaging in stay-at-home behaviors. Specifically, compared with Americans, Chinese people are more likely to stay at home during peaks in COVID-19.*


### 1.2. The Mediating Role of Health Beliefs

Health beliefs primarily comprise two aspects: threat perception and behavioral evaluation [19,20]. Specific to the current study, threat perception involves both “perceived susceptibility”, belief about the likelihood of contracting COVID-19, and “perceived severity”, belief about the seriousness of contracting COVID-19. Behavioral evaluation involves both “perceived benefits” that are associated with beliefs about the potential gains of staying at home and “perceived barriers” that are associated with beliefs about potential obstacles to staying at home. Based on the health belief model [8], individual factors such as nationality may impact the extent to which people perceive COVID-19 as a health threat and evaluate stay-at-home behaviors as a health-promoting action, which further influences the frequency with which people engage in stay-at-home behaviors during the pandemic.

With respect to the first stage of the link, nationality may be associated with health beliefs. In terms of perceived susceptibility and severity, collectivism is often associated with an extended family structure while individualism is often associated with a nuclear family structure [13]. Compared with Americans shaped by individualism, Chinese people who are influenced by collectivism may maintain closer social relationships and exhibit more frequent physical connections with in-group members, which may increase the risk of being exposed to the coronavirus and the perception of the likelihood of being infected [21]. Furthermore, collectivists like Chinese tend to be concerned about the consequences of their actions on others [15], leading to the belief that disease infection is not a simple matter of individual consequences but might bring substantial harm to a group [21,22]. This belief may manifest as concerns about infecting others, especially parents and children, and result in overestimation of the seriousness of disease infection. Several studies supported these notions by demonstrating that collectivism/individualism was positively/negatively associated with perceived susceptibility and severity of a disease [23,24]. Thus, it appears that compared with Americans, Chinese people might perceive higher levels of susceptibility and severity in contracting COVID-19.

Regarding perceived benefits and barriers, Chinese people shaped by collectivism may share similar social norms and internalize the group’s goals and values [13]; thus, they are more likely to recognize and identify the potential benefits of the actions recommended by the group they belong to. Accordingly, the Chinese are more likely to perceive the benefits of engaging in stay-at-home behaviors, especially under enforcement and endorsement by the government. In contrast, Americans with high individualism may give personal interests a higher priority over group interests and emphasize the importance of personal autonomy [13]; thus, they may find it difficult to restrict individual movement or to confine themselves at home over a long period [2]. Thus, it seems that compared with Chinese people, Americans may be more likely to perceive the barriers associated with stay-at-home behaviors.

The effects of health beliefs on disease-preventive behaviors have been widely validated in numerous empirical studies [22,24,25,26]. For instance, Yoshitake, Omori, Sugawara, Akishinonomiya, Shimada, and Iannello [22] found that perceived susceptibility to tuberculosis positively predicted the frequency of engaging in preventive behaviors such as wearing surgical masks in public. Similarly, Alsulaiman and Rentner [26] found that study participants in Saudi Arabia who had higher levels of perceived susceptibility and severity regarding the coronavirus Middle East respiratory syndrome, and higher levels of perceived benefits and lower levels of perceived barriers regarding preventive behaviors, were more likely to implement disease-preventive behaviors recommended by the government. In terms of the COVID-19 pandemic, it has also been found that perceived susceptibility, perceived severity, and perceived benefits positively predicted, and perceived barriers negatively predicted, the frequency of engaging in preventive behaviors such as staying at home unless it is for essential activities [24]. Therefore, we propose the following hypothesis (see Figure 1):

**Hypothesis** **2** **(H2).**
*Health beliefs mediate the association between nationality and engaging in stay-at-home behaviors. Specifically, compared with Americans, Chinese people perceive higher susceptibility, higher severity, more benefits, and fewer barriers, leading to increased stay-at-home behaviors during peaks in COVID-19.*


### 1.3. The Mediating Role of Behavioral Intention

Behavioral intention is assumed to capture the motivational factors that influence behaviors [27] and has been considered the central factor in linking internal cognitive beliefs and actual behaviors in the theory of reasoned action [10] and the technology acceptance model [11]. The former model assumes that attitude and subjective norm affect behavioral intention and the latter model assumes that perceived usefulness and ease to use affect behavioral intention. In turn, behavioral intention influences actual behaviors [28,29]. In the context of the COVID-19 pandemic, when people perceive high levels of health threats, such as a high risk of being infected and uncontrollable consequences after being infected, these beliefs may be associated with negative attitudes toward disease infection and positive attitudes toward preventive behaviors, which has the potential to enhance preventive-related intention [30]. Similarly, when people believe that staying at home is useful to prevent infection and is easy to execute, they are more likely to generate the intention to stay at home for disease prevention. In short, the perception of a high health threat and positive behavioral evaluation may influence stay-at-home intention during the pandemic.

Ample evidence has shown the convergence between behavioral intention and actual behaviors [31]. For instance, a meta-analysis of meta-analyses regarding various behavioral domains showed that intention explained 28% of the variance of enacting behaviors on average [32]. More importantly, a meta-analysis of experimental evidence found that the changes in intention led to changes in behaviors, showing that interventions to produce large changes in intention caused medium changes in actual behaviors [33]. In terms of the COVID-19 pandemic, empirical evidence has also indicated that the stronger the preventive intention, the higher the frequency of actual preventive behaviors [30]. Therefore, we propose the following hypothesis (see Figure 1):

**Hypothesis** **3** **(H3).**
*Behavioral intention mediates the association between health beliefs and engaging in stay-at-home behaviors. Specifically, perceived susceptibility, perceived severity, and perceived benefits are positively associated with behavioral intention, and perceived barriers are negatively associated with behavioral intention, leading to frequent stay-at-home behaviors during peaks in COVID-19.*


### 1.4. The Mediation Model

As a type of demographic characteristic, nationality associated with cultural orientation may influence perceptions about health threats and evaluations on health behaviors, which facilitate changes in behavioral intention and subsequent increases in the frequency of preventive behaviors according to the health belief model [8,9], the theory of reasoned action [10], and the technology acceptance model [11]. Specifically, compared with Americans who are oriented toward individualism [13], the Chinese who are oriented toward collectivism may have closer interpersonal relationships and remain committed to social norms, leading to the perception of higher susceptibility and severity of infection with COVID-19 and more benefits and fewer barriers to engaging in stay-at-home behaviors. These health beliefs may increase the intention to prevent infection, which further leads to frequent preventive behaviors like staying at home. Therefore, we propose the following hypothesis (see Figure 1):

**Hypothesis** **4** **(H4).**
*Nationality is indirectly associated with engaging in stay-at-home behaviors through the multiple mediating effects of health beliefs and behavioral intention. Specifically, compared with Americans, Chinese people perceive higher susceptibility, higher severity, more benefits, and fewer barriers, which increase the intention to stay at home, leading to frequent stay-at-home behaviors during peaks in COVID-19.*


## 2. Methods

### 2.1. Participants and Procedures

This research was conducted in China and the United States because the former is a typical collectivist country while the latter is a typical individualistic country, as mentioned above. We administered self-report questionnaires on online survey platforms (i.e., Wenjuanxing in China and Qualtrics in the USA) and participants voluntarily participated in the online investigation. During May and June 2020, a total of 790 college students, including 498 in China and 292 in the United States, completed the online questionnaires. These participants comprised 231 (29.2%) males, 553 (70.0%) females, and 6 (0.8%) students reporting non-binary gender, and they had an age average of 20.93 years (*SD* = 2.89), with a range from 17 to 30.

This research obtained approval from the Academic Ethics Committee of the Faculty of Psychology at Beijing Normal University and Fordham University. The original English version of the questionnaires was used in American participants; the Chinese version used in Chinese participants was translated by two Chinese psychology doctoral students and two native English speakers, following the standard backward and forward approach, which to some extent ensured the measure equivalence in different cultures. Before participants began to answer the self-report questionnaires, they were provided with an informed consent form that noted their voluntary participation in the investigation and their right to opt out of the online survey at any time. Furthermore, they were informed that their responses would be kept confidential and used only for academic research purposes. Completing the questionnaires took approximately 10 min, and participants were compensated with an incentive (e.g., a special postcard or a gift card through raffles).

### 2.2. Measures

#### 2.2.1. Perceived Susceptibility

Perceived susceptibility of COVID-19 was assessed by a questionnaire based on the health belief model that was adapted to fit the context of the COVID-19 pandemic [34]. The questionnaire contained four items (see Table 1) and used a 5-point Likert scale, ranging from 1 (strongly disagree) to 5 (strongly agree), with higher scores indicating a higher level of perceived susceptibility of COVID-19. This questionnaire had good reliability and validity in the present study (Cronbach’s α = 0.86, the composite reliability (CR) = 0.87, the average variance extracted (AVE) = 0.64).

#### 2.2.2. Perceived Severity

Perceived severity of COVID-19 was assessed by a similar adapted questionnaire [34]. It contained four items (see Table 1) and used a 5-point Likert scale, ranging from 1 (strongly disagree) to 5 (strongly agree), with higher scores indicating a higher level of perceived severity of COVID-19. This questionnaire had good reliability and validity in this study (Cronbach’s α = 0.79, CR = 0.80, AVE = 0.51).

#### 2.2.3. Perceived Benefits

Perceived benefits of staying at home were measured by a questionnaire adapted from a measure that was used to assess the perception of the outcomes (including benefits and barriers) of implementing SARS-preventive behaviors [12]. The questionnaire contained four items (see Table 1) and used a 5-point Likert scale, ranging from 1 (strongly disagree) to 5 (strongly agree), with higher scores indicating a higher level of perceived benefits. This questionnaire had acceptable reliability and validity in this study (Cronbach’s α = 0.77, CR = 0.77, AVE = 0.46).

#### 2.2.4. Perceived Barriers

Perceived barriers to staying at home were measured by a similar adapted questionnaire [12]. It contained five items (see Table 1) and used a 5-point Likert scale, ranging from 1 (strongly disagree) to 5 (strongly agree), with higher scores indicating a higher level of perceived barriers. This questionnaire had satisfactory reliability and validity in this study (Cronbach’s α = 0.91, CR = 0.91, AVE = 0.68).

#### 2.2.5. Behavioral Intention

The questionnaire aiming to assess the intention to stay at home was adapted from the research of Davis [35], and Moon and Kim [36]. It contained three items (see Table 1) and used a 5-point Likert scale, ranging from 1 (strongly disagree) to 5 (strongly agree), with higher scores indicating a higher level of intention to stay at home during peaks in COVID-19. This questionnaire had good reliability and validity in this study (Cronbach’s α = 0.87, CR = 0.88, AVE = 0.70).

#### 2.2.6. Actual Behaviors

The questionnaire aiming to assess actual stay-at-home behaviors was adapted from the research of Moon and Kim [36], and Lu, Zhou, and Wang [28]. It contained three items (see Table 1) and each item was rated on a 7-point Likert scale. After the scores were reversed, higher scores indicated higher frequencies of staying at home during the peak of COVID-19. This questionnaire had good reliability and validity in this study (Cronbach’s α = 0.87, CR = 0.87, AVE = 0.70).

### 2.3. Data Analyses

First, we employed an exploratory factor analysis (EFA) to extract factors and employed a confirmatory factor analysis (CFA) to examine the reliability and validity of all constructs, which helped to ensure the robustness of the measurement model. Second, means, standard deviations, and Pearson correlations among the variables were calculated using SPSS 19.0. Third, the structural model was tested by structural equation modeling (SEM) using Mplus 7.1 [37]. As suggested by Wen et al. [38], whether a model was acceptable was evaluated by the model fit indicators, including the chi-square values (χ^2^), the comparative fit index (CFI), the Tucker–Lewis fit index (TLI), the root mean square error of approximation (RMSEA), and the standardized root mean square residual (SRMR). When CFI and TLI are at 0.90 or above and the RMSEA and SRMR are at 0.08 or lower, the model can be considered an acceptable model.

## 3. Results

### 3.1. The Measurement Model

To examine the reliability and validity of the six constructs, EFA was used to explore the dimensionality of all items. The results showed that Bartlett’s test of sphericity was significant (*p* < 0.001) and the Kaiser–Meyer–Olkin measure of sampling adequacy was 0.84. These findings met the prerequisite of EFA. As displayed in Table 1, six factors were extracted from the items accounting for 63.3% of the variance and were consistent with the hypothetical constructs (i.e., perceived susceptibility, perceived severity, perceived benefits, perceived barriers, behavioral intention, and actual behaviors).

Furthermore, CFA was used to further examine the measurement model. Perceived susceptibility, perceived severity, perceived benefits, perceived barriers, behavioral intention, and actual behaviors were loaded on four, four, four, five, three, and three items, respectively. Not only an overall CFA but also two separate CFAs for the Chinese and American participants were conducted. As displayed in Table 2 and Figure 2, the model results showed good model fits and all the loadings were significant (*p* < 0.001). Additional tests on the measurement equivalence according to the ratio of Δχ^2^ and Δdf [39] showed that model 3 and model 2 had no significant differences (Δχ^2^/Δdf = 6.35 < 27.59, the cutoff of chi-square test with df = 17 was at the 0.05 level of significance). Similarly, model 4 and model 3 had no significant differences (Δχ^2^/Δdf = 14.48 < 22.36, the cutoff of chi-square test with df = 13 was at the 0.05 level of significance). These findings indicated that the measurement model had strong invariance for Chinese and American participants.

### 3.2. Descriptive Statistics and Correlations

Means, standard deviations, and Pearson correlations are presented in Table 3. Gender was correlated with other variables except for perceived severity, and age was correlated with other variables except for behavioral intention. These correlations indicated that gender and age should be considered as covariates in the next step of analysis. Regarding the correlations among the main variables, nationality was correlated with other variables except for behavioral intention. Perceived susceptibility, perceived severity, perceived benefits, perceived barriers, behavioral intention, and actual behaviors were correlated with each other (0.13 < |*r*| < 0.52) except for non-significant associations between perceived susceptibility and perceived benefits, between perceived susceptibility and behavioral intention, and between perceived benefits and perceived barriers. It is worth noting that the square roots of the AVEs (see the diagonal elements in bold in Table 2) were greater than inter-construct correlations, which indicated good discriminant validity among these constructs [40].

### 3.3. The Structural Model

To examine the hypothetical model, SEM was used to test the mediating effects of perceived susceptibility, perceived severity, perceived benefits, perceived barriers, and behavioral intention in the association between nationality and actual stay-at-home behaviors. Because nationality was a binary variable, dummy coding was used (0 = USA; 1 = China). The mediation results showed a good model fit, χ^2^/df = 4.12, CFI = 0.92, TLI = 0.91, RMSEA = 0.06, SRMR = 0.07. As shown in Figure 3, after the effects of gender and age were controlled, nationality not only directly predicted actual behaviors but also significantly predicted perceived susceptibility, perceived severity, and perceived barriers (but not perceived benefits). Furthermore, although four types of health beliefs did not directly predict actual stay-at-home behaviors, they did exert indirect effects through the mediating effects of behavioral intention.

To further examine the significance of the indirect effects according to the suggestions of Cheung [37], a bias-corrected bootstrap test derived with 1000 samples was used. It was known that the 95% confidence interval of the mediating path without including zero indicated the significance. As illustrated in Table 4, all the simple mediating effects of health beliefs (i.e., perceived susceptibility, perceived severity, perceived benefits, and perceived barriers) were not significant, but the multiple mediating effects of perceived susceptibility → behavioral intention, perceived severity → behavioral intention, and perceived barriers → behavioral intention (but not perceived benefits → behavioral intention) were significant in the association between nationality and actual behaviors. To further compare the effect sizes of different health beliefs, Wald’s tests were conducted. The effect size of perceived barriers → behavioral intention was greater than that of perceived susceptibility → behavioral intention (*t* = 2.13, *p* < 0.05) and perceived severity → behavioral intention (*t* = 2.59, *p* < 0.01). Altogether, these findings indicated that American participants perceived high levels of susceptibility, Chinese participants perceived high levels of severity, especially few barriers, which in turn increased the intention to stay at home and facilitated actual behaviors during peaks in COVID-19.

## 4. Discussion

This study was the first of its kind to explore the differences between Chinese and American college students in their self-reported stay-at-home behaviors during peaks in COVID-19. Based on the health belief model [8,9], combined with the theory of reasoned action [10] and the technology acceptance model [11], health beliefs and behavioral intention were suggested as a link to unravel the potential mechanisms underlying the national differences. The results revealed that Chinese college students reported more frequent stay-at-home behaviors than American students, concurring with the extant findings [15]. Furthermore, health beliefs (including perceived susceptibility, severity, and barriers, but not including benefits) and behavioral intention played multiple mediating roles in the association between nationality and stay-at-home behaviors, respectively. Notably, the effect via perceived barriers → behavioral intention was stronger than the effects via perceived susceptibility and severity → behavioral intention. In other words, American college students perceived high levels of susceptibility whereas Chinese college students perceived high levels of severity, especially fewer barriers, leading to enhanced behavioral intention and frequent stay-at-home behaviors during the peak of COVID-19.

### 4.1. The Mediating Roles of Perceived Susceptibility and Behavioral Intention

The results showed that compared with American college students, Chinese college students reported lower levels of perceived susceptibility, although it was believed that perceived susceptibility increased stay-at-home intention and actual behaviors during the pandemic, contrary to the hypothesis. This may be due to the paradoxical influence of collectivist cultures on reactivity to threatening diseases [23]. On one hand, collectivism encourages people to maintain close social contacts, which may increase disease-related risk perceptions [21,24]. On the other hand, collectivism serves as a psychological buffer against disease infection [23]. In the context of this pandemic, Chinese people, including college students, may worry about being infected partially due to intensive social interactions, but they tend to share resources and stay united to protect their groups against diseases [41]. To a larger extent, these processes may strengthen Chinese college students’ faith in overcoming the pandemic and help with alleviating the obsessive concerns about infection risks, thereby exhibiting a relatively lower level of susceptibility in comparison to American college students. Despite this, future studies are warranted to further explore the complicated relations between nations and vulnerable responses to infectious diseases.

### 4.2. The Mediating Roles of Perceived Severity and Behavioral Intention

The results suggested that compared with American college students, Chinese college students reported higher levels of perceived severity, which led to increased stay-at-home behavioral intention and more frequent actual stay-at-home behaviors during the pandemic, supporting the hypothesis. Moreover, this finding was consistent with previous research [21,24]. The extended Chinese family structure often includes not only parents and siblings but also grandparents, uncles, and aunts with whom they interact intensively [13]. Given that COVID-19 transmission occurs mainly through direct, indirect, or close contact with infected secretions [1], such family circumstances are likely to facilitate the belief that disease infection not only threatens their health but also poses a risk to the belonged family’s and group’s security [21] due in part to that collectivists tend to be concerned about the influence of their behaviors on others [15]. Accordingly, Chinese students living in extended family households have the potential to overemphasize the seriousness of getting infected. In this sense, their perception of high severity motivates preventive plans such as staying at home, and intentions to practice those plans, following by frequent preventive behaviors during peaks in COVID-19 [30].

### 4.3. The Mediating Roles of Perceived Benefits and Behavioral Intention

The results showed that nationality was not associated with perceived benefits although perceived benefits increased stay-at-home intention and actual behaviors in these college students, not supporting the mediation hypothesis. This finding indicated that there was no difference in perceived benefits of stay-at-home behaviors between Chinese and American college students. One plausible explanation may be the general impact of public knowledge about COVID-19 [42]. Recent surveys indicate an increase in social media usage among both Chinese and Americans, including college students, to obtain pandemic-related information [43,44]. During the COVID-19 outbreak, public media have broadcast the substantial efforts of governments (including those of China and the United States) to provide reliable information such as usefulness and effectiveness about necessary preventive behaviors [45]. Understandably, Chinese and American college students might be directly or indirectly exposed to the prevention information and influenced by such information; thus, they are exhibiting no difference in perceiving the benefits of enacting stay-at-home behaviors during the pandemic.

### 4.4. The Mediating Roles of Perceived Barriers and Behavioral Intention

The results showed that compared with American college students, Chinese college students reported lower levels of perceived barriers, which led to increased stay-at-home behavioral intention and more frequent stay-at-home behaviors during the pandemic, supporting the hypothesis. In particular, American college students may be shaped by individualist cultures in which the interests of the individual prevail over the interests of the group [13]; thus, they are more likely to act autonomously and less likely to sacrifice their personal desires for the benefit of others. In the face of lockdown policies, individualists may believe that restricting individual movement will interfere with their daily routines and thus are less likely to comply with recommendations to confine themselves [2]. Stated differently, American college students are more likely to perceive more physical and psychological obstacles to staying at home. Influenced by the perceived barriers, these students might be less likely to generate stay-at-home intention and perform actual preventive behaviors. In contrast, Chinese college students are more likely to perceive fewer barriers, leading to an increase in stay-at-home intention and actual behaviors during peaks in COVID-19.

### 4.5. Limitations and Future Directions

Several limitations to this study should be noted. First, bias might occur when measures were used in different cultures [46] although additional tests on the measure equivalence showed strong invariance. Future research could conduct pilot studies before formal surveys to further enhance the reliability and validity of measures when used in different cultures. Second, this study recruited only college students from one typical collectivist country and one typical individualistic country; thus, the generalization to other populations and other countries should be made with caution. Future studies could investigate more participants in more countries oriented in corresponding cultures. Third, this study focused only on a specific preventive behavior (i.e., staying at home), which might provide limited contributions to halting the COVID-19 pandemic. Future research could consider other effective preventive measures (e.g., wearing facial masks) for a comprehensive understanding of overcoming this global challenge.

### 4.6. Theoretical and Practical Implications

From a theoretical perspective, this study was the first of its kind to explore the cross-country differences in COVID-19-related preventive behaviors, showing that Chinese college students reported engaging in more frequent stay-at-home behaviors in comparison to American students. Furthermore, this study identified the mediating effects of health beliefs and behavioral intention underlying the cross-country differences. That is, nationality was positively associated with perceived severity and negatively associated with perceived susceptibility and barriers, which had indirect effects on the frequency of stay-at-home behaviors completely through the mediating role of behavioral intention. These findings extend the health belief model, the theory of reasoned action, and the technology acceptance model into the context of behavioral differences in taking preventive measures and offer valuable insights for disease prevention and health promotion.

From a practical perspective, the results revealed that American college students with high individualism perceived high levels of susceptibility of disease infection, leading to increased preventive intention and actual behaviors. Unlike American college students, Chinese students with high collectivism perceived high levels of severity of disease infection, especially few barriers to enacting preventive behaviors, leading to increased behavioral intention and frequent behaviors. These findings suggest that policymakers follow the example of the US government and raise public awareness regarding individual vulnerability, such as by publicizing the widespread transmission of the coronavirus, in favor of increasing the perception about infection risks. Similarly, they could also learn from the experience of the Chinese government, which in a collective manner, has encouraged people to formulate a shared sense of responsibility, emphasized group interests, and advocated actions for the common good [2]. Furthermore, all traditional and new social media should continue to broadcast reliable COVID-19 preventive information to highlight the explicit benefits and positive outcomes, particularly for both individuals and groups, when complying with public health orders and implementing preventive behaviors. More importantly, governments should utilize strategies that assist in overcoming physical and psychological barriers to practicing preventive measures, such that the Chinese government organized primary facilities and volunteer groups to help residents who stayed at home with essential activities like buying living and medical supplies. Additionally, public health experts could instruct people on how to make self-isolation decisions, such as using decision aids (e.g., decision trees and planning tools), to enhance stay-at-home intention [5]. Overall, policymakers could take measures based on the above processes to influence health beliefs and behavioral intention, which may not only facilitate disease prevention at the individual level, but also contribute to mitigating the health crisis at the global level.

## 5. Conclusions

This study focused on an effective preventive behavior—staying at home—to avoid contracting COVID-19. Cross-country differences between China and the United States in complying with this behavior have been manifested during peaks in this pandemic. Internal cognitive processes, such as health beliefs and behavioral intention, were suggested to explicate the cross-country differences. The structural equation modeling results showed that the links of health beliefs (i.e., perceived susceptibility, severity, and barriers) and behavioral intention significantly mediated the association between nationality and stay-at-home behaviors. Specifically, American college students perceived high susceptibility of contracting COVID-19 whereas Chinese college students perceived high severity of contracting COVID-19, especially few barriers to staying at home, which further led to increased behavioral intention and frequent actual behaviors. Overall, these findings provide a scientific basis for the explanation mechanism of the cross-country differences in preventive behaviors, which offers practical insights to understanding how to mitigate the global COVID-19 pandemic.

## Figures and Tables

**Figure 1 ijerph-18-02104-f001:**
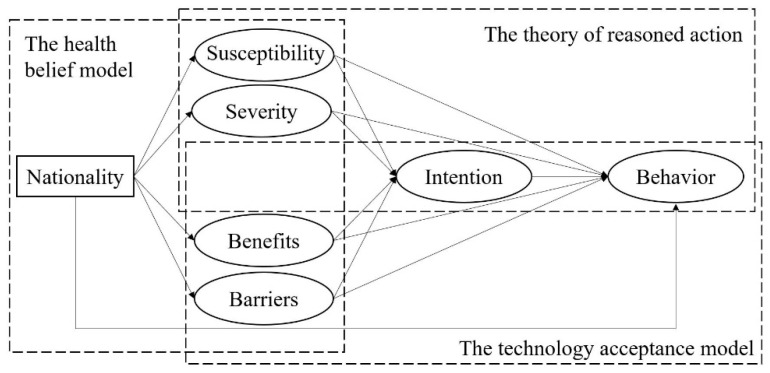
The conceptual model of the national differences in stay-at-home behaviors.

**Figure 2 ijerph-18-02104-f002:**
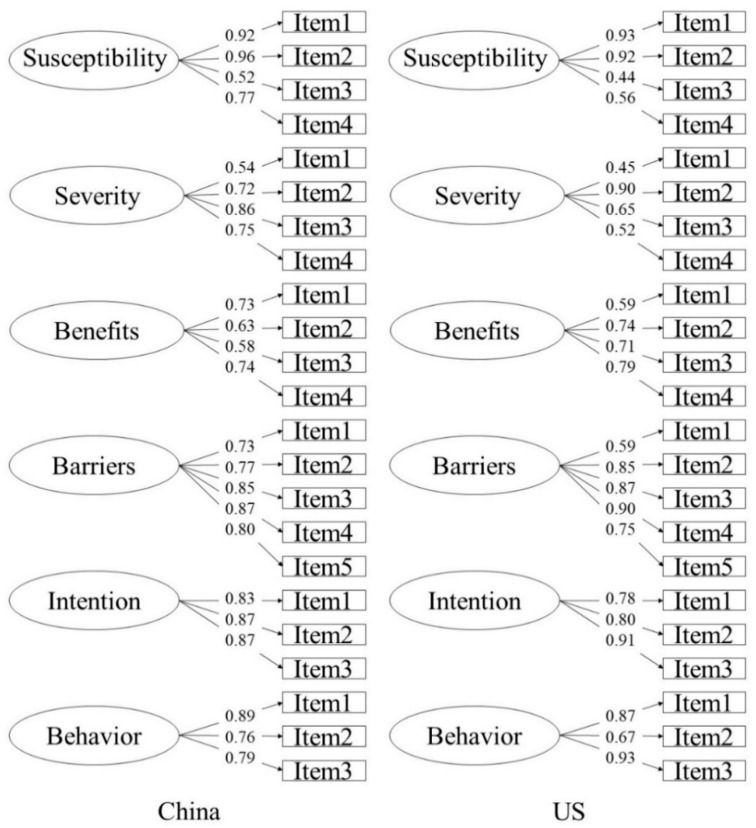
The confirmatory factor analysis (CFA) results of the measurement model in Chinese and US participants.

**Figure 3 ijerph-18-02104-f003:**
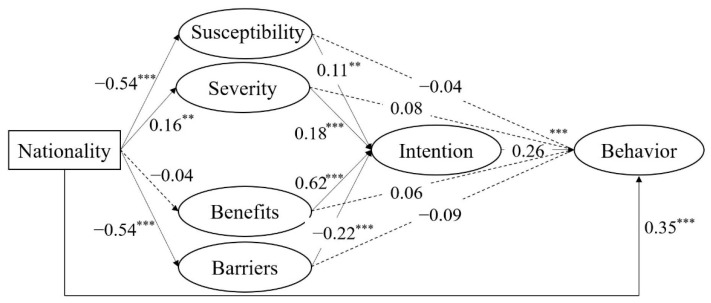
The model of health beliefs and behavioral intention between nationality and engaging in stay-at-home behaviors. Note. The solid/dotted lines indicate significance/non-significance. All coefficient estimates are completely standardized. Nationality (0 = USA, 1 = China). * *p* < 0.05, ** *p* < 0.01, *** *p* < 0.001.

**Table 1 ijerph-18-02104-t001:** The exploratory factor analysis results of the measures.

During Peaks in the COVID-19 Pandemic,	Factor 1	Factor 2	Factor 3	Factor 4	Factor 5	Factor 6
1. My chances of getting COVID-19 are great.	0.94					
2. There is a good possibility that I will get COVID-19.	0.93					
3. I worry a lot about getting COVID-19.	0.53					
4. I am more likely than the average person to get COVID-19.	0.73					
5. COVID-19 is a hopeless disease.		0.50				
6. Problems I would experience from COVID-19 would last a long time.		0.72				
7. Getting COVID-19 would result in serious consequences.		0.84				
8. If I got COVID-19, my life would change.		0.76				
9. Staying at home prevents me from getting COVID-19.			0.69			
10. If I do not stay at home, it is more likely that I will get COVID-19.			0.60			
11. If I stay at home, I would become less anxious about getting COVID-19.			0.64			
12. Staying at home can help me to stay in a healthy condition.			0.71			
13. Staying at home causes me inconvenience.				0.75		
14. Staying at home interferes with my activities.				0.86		
15. If I stay at home, I will have to break my usual life habits.				0.84		
16. If I stay at home, my daily schedule will be disrupted.				0.83		
17. In order to stay at home, I have to give up quite a bit.				0.82		
18. I will always stay at home except for essential activities.					0.82	
19. I will recommend others to stay at home.					0.78	
20. I will continue staying at home.					0.88	
21. How many times did you go outside each week?						0.90
22. How many hours did you go outside each week?						0.74
23. How frequently did you go outside?						0.85
**Percentage of variance accounted for (%)**	**8.96**	**8.11**	**16.19**	**21.44**	**2.85**	**5.74**
**Cronbach α for subscale**	**0.86**	**0.79**	**0.77**	**0.91**	**0.87**	**0.87**
**Composite reliability (CR)**	**0.87**	**0.80**	**0.77**	**0.91**	**0.88**	**0.87**
**Average variance extracted (AVE)**	**0.64**	**0.51**	**0.46**	**0.68**	**0.70**	**0.70**

Note. Extraction method: principal axis factoring; rotation method: oblimin with Kaiser normalization.

**Table 2 ijerph-18-02104-t002:** Tests on the measurement equivalence between Chinese and US participants.

Models	Interpretations	χ^2^	df	CFI	TLI	RMSEA	SRMR	Δχ^2^	Δdf
Model 1a	China	683.50	215	0.93	0.91	0.07	0.06	–	–
Model 1b	USA	485.75	213	0.92	0.90	0.07	0.07	–	–
Model 2: configural invariance	same indicators	1066.79	428	0.93	0.92	0.06	0.07	–	–
Model 3: metric invariance	same loadings	1174.78	445	0.92	0.91	0.06	0.08	107.99	17
Model 4: scalar invariance	same intercepts	1363.08	458	0.91	0.90	0.07	0.08	188.30	13

**Table 3 ijerph-18-02104-t003:** Means, standard deviations, and correlations.

Variables	1	2	3	4	5	6	7	8	9
1 gender	–								
2 age	0.30 ***	–							
3 nationality	−0.29 ***	−0.59 ***	–						
4 susceptibility	0.18 ***	0.33 ***	−0.51 ***	**0.80**					
5 severity	−0.05	−0.09 *	0.17 ***	0.13 ***	**0.72**				
6 benefits	0.07 *	0.11 **	−0.09 **	−0.03	0.15 ***	**0.68**			
7 barriers	0.07 *	0.25 ***	−0.45 ***	0.29 ***	0.04	−0.02	**0.82**		
8 intention	0.09 *	0.05	0.04	0.04	0.22 ***	0.52 ***	−0.17 ***	**0.84**	
9 behavior	−0.11 **	−0.25 ***	0.40 ***	−0.20 ***	0.19 ***	0.18 ***	−0.28 ***	0.32 ***	**0.83**
*M*	–	20.83	–	2.31	3.25	3.97	3.24	4.00	5.45
*SD*	–	2.89	–	0.91	0.83	0.71	1.03	0.80	1.15

Note. Diagonal elements in bold are the square roots of the average variance extracted (AVE). Gender (0 = male, 1 = female). Nationality (0 = USA, 1 = China). * *p* < 0.05, ** *p* < 0.01, *** *p* < 0.001.

**Table 4 ijerph-18-02104-t004:** Bias-corrected bootstrap tests on the direct and indirect effects.

Paths	Standardized (β)	95% CI		Significance
		Low	High	
Nationality → Behavior	0.347	0.237	0.456	√
Nationality → Susceptibility → Behavior	0.020	−0.029	0.068	×
Nationality → Severity → Behavior	0.012	−0.004	0.028	×
Nationality → Benefits → Behavior	−0.002	−0.013	0.008	×
Nationality → Barriers → Behavior	0.047	−0.005	0.099	×
Nationality → Susceptibility → Intention → Behavior	−0.015	−0.030	−0.001	√
Nationality → Severity → Intention → Behavior	0.007	0.001	0.014	√
Nationality → Benefits → Intention → Behavior	−0.006	−0.023	0.011	×
Nationality → Barriers → Intention → Behavior	0.032	0.012	0.052	√

## Data Availability

The data presented in this study are available on request from the corresponding author. The data are not publicly available due to participants’ privacy.

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
