# Peer review of "Cross-Country Differences in Stay-at-Home Behaviors during Peaks in the COVID-19 Pandemic in China and the United States: The Roles of Health Beliefs and Behavioral Intention"

_ijerph, 2021, doi:10.3390/ijerph18042104_

Round 1
Reviewer 1 Report
This study aimed to investigate the cross-country differences between China and the United States in disease-preventive behaviors and its potential mechanisms from the perspective of cognitive beliefs and behavioral intention, which lend support for mitigating the global COVID-19 pandemic.
The paper is organized relatively well, the argument is logical and well-argued, and an overview of some key literature is provided. Despite these strengths, the manuscript has some limitations:
(1). What is the importance of selecting the two countries that Authors did to compare and contrast? Why these two countries? Was it a simple convenience? Or is there a more important reason? If so, Authors should clearly let the reader know it.
(2). Line: 192-196 Authors described sample but fails to describe sampling procedures.
(3). Whether the research tool has been culturally adapted? The authors should provide information about it.
Moreover, whether pilot studies have been carried out using that tool after standard back-translation procedures (line 201-202)?
The main problem of such kind of approach is construct bias which appears when a measured construct is different in culture (van de Vijver, 2011): Bias and real differences.
Next one is methods bias, which consists of sample bias, items bias and administrator bias. The sample bias in cross-culture research is involved in the problem of incompatibility of simple caused by individuals and groups' cultural diversity.
(4). Limitations of this study are missing.
Overall, an interesting and thoughtful paper.
Author Response
This study aimed to investigate the cross-country differences between China and the United States in disease-preventive behaviors and its potential mechanisms from the perspective of cognitive beliefs and behavioral intention, which lend support for mitigating the global COVID-19 pandemic.
The paper is organized relatively well, the argument is logical and well-argued, and an overview of some key literature is provided. Despite these strengths, the manuscript has some limitations:
1) What is the importance of selecting the two countries that Authors did to compare and contrast? Why these two countries? Was it a simple convenience? Or is there a more important reason? If so, Authors should clearly let the reader know it.
Response:
Thanks a lot for your suggestions.
1) We have strengthened the rationale for selecting these two countries because China is categorized as a typical collectivist country while the United States is categorized as a typical individualistic country according to the perspective of Hofstede et al. (2010). (see Line 174-175)
2) We have added this limitation – focusing on only these two countries and the generalization of the findings – in the Discussion Section. (see Line 393-396)
References:
Hofstede, G., Hofstede, G. J., & Minkov, M. (2010). Cultures and organizations: Software of the mind (3rd ed.). McGraw-Hill.
2) Line: 192-196 Authors described sample but fails to describe sampling procedures.
Response:
Thanks for your comments. We have added more details about how we recruited participants as followed (see Line 174-179):
This research was conducted in China and the United States because the former is a typical collectivist country while the latter is a typical individualistic country mentioned above. We administered self-report questionnaires on online survey platforms (i.e., Wenjuanxing in China and Qualtrics in the U.S.) and participants voluntarily participated in the online investigation. During May and June 2020, a total of 790 college students, including 498 in China and 292 in the United States, completed the online questionnaires.
3) Whether the research tool has been culturally adapted? The authors should provide information about it.
Moreover, whether pilot studies have been carried out using that tool after standard back-translation procedures (line 201-202)?
The main problem of such kind of approach is construct bias which appears when a measured construct is different in culture (van de Vijver, 2011): Bias and real differences.
Next one is methods bias, which consists of sample bias, items bias and administrator bias. The sample bias in cross-culture research is involved in the problem of incompatibility of simple caused by individuals and groups' cultural diversity.
Response:
Thank you very much for your insightful comments.
1) From a procedural perspective, first, we adapted the original English version of measures to fit the context of the COVID-19 pandemic. Next, we invited two Chinese psychology doctoral students and two native English speakers to translate the English version into a Chinese version, following the standard backward and forward approach. These processes may help to increase the measurement equivalence in different cultures. (see Line 183-187)
2) From a statistical perspective, except for separate confirmatory factor analyses on Chinese and American participants, we added additional tests on the measurement equivalence and compared different models (i.e., Model 2: configural invariance, Model 3: metric invariance, and Model 4: scalar invariance), the results showed the measurement model had strong invariance and indicated that there was no serious bias of the measures between Chinese and American participants in this study. (see Line 259-264 and Table 2)
3) Despite this, we have noted these limitations – measure bias and sample bias – in the Discussion Section as followed (see Line 390-396):
First, bias might occur when measures were used in different cultures (He & van de Vijver, 2012) although additional tests on the measure equivalence showed strong invariance. Future research could conduct pilot studies before formal surveys to further enhance the reliability and validity of measures when used in different cultures. Second, this study recruited only college students from one typical collectivist country and one typical individualistic country; thus, the generalization to other populations and other countries should be made with caution. Future studies could investigate more participants in more countries oriented in corresponding cultures.
References:
He, J., & van de Vijver, F. (2012). Bias and equivalence in cross-cultural research. Online Readings in Psychology and Culture, 2(2). https://doi.org/10.9707/2307-0919.1111
4) Limitations of this study are missing.
Response:
Thanks for your comments. We have added three limitations of this study in the Discussion Section as followed (see Line 390-400):
Several limitations to this study should be noted. First, bias might occur when measures were used in different cultures (He & van de Vijver, 2012) although additional tests on the measure equivalence showed strong invariance. Future research could conduct pilot studies before formal surveys to further enhance the reliability and validity of measures when used in different cultures. Second, this study recruited only college students from one typical collectivist country and one typical individualistic country; thus, the generalization to other populations and other countries should be made with caution. Future studies could investigate more participants in more countries oriented in corresponding cultures. Third, this study focused only on a specific preventive behavior (i.e., staying at home), which might provide limited contributions to halting the COVID-19 pandemic. Future research could consider other effective preventive measures (e.g., wearing facial masks) for a comprehensive understanding of overcoming this global challenge.
References:
He, J., & van de Vijver, F. (2012). Bias and equivalence in cross-cultural research. Online Readings in Psychology and Culture, 2(2). https://doi.org/10.9707/2307-0919.1111
Overall, an interesting and thoughtful paper.
Reviewer 2 Report
It is a good paper, coherent and well organized.
The comparison between the behaviors of university students of different nationalities in the face of the health emergency experienced is of great interest.
I have doubts about the generalizability of the results, especially since the results based on a sample of 790 students (China and the United States) are generalized to understand cross-country differences.
Therefore, it would be convenient:
1) Include the measurement error in the sample analysis, indicating the population from which the sample is drawn. Add a comment on the possibilities or limitations of the generalization of the results.
2) Adjust the discussion and conclusions taking into account this (understandable) sampling limitation.
3) Add a section of conclusions
Author Response
It is a good paper, coherent and well organized.
The comparison between the behaviors of university students of different nationalities in the face of the health emergency experienced is of great interest.
I have doubts about the generalizability of the results, especially since the results based on a sample of 790 students (China and the United States) are generalized to understand cross-country differences.
Therefore, it would be convenient:
1) Include the measurement error in the sample analysis, indicating the population from which the sample is drawn. Add a comment on the possibilities or limitations of the generalization of the results.
Response:
Thanks a lot for your concrete suggestions. We have added this limitation in the Discussion Section as followed (see Line 393-396):
This study recruited only college students from one typical collectivist country and one typical individualistic country; thus, the generalization to other populations and other countries should be made with caution. Future studies could investigate more participants in more countries oriented in corresponding cultures.
2) Adjust the discussion and conclusions taking into account this (understandable) sampling limitation.
Response:
Thanks for your suggestions. We have revised the Discussion Section and interpreted the findings that were limited in Chinese and American college students. In addition, we have noted this limitation in the Discussion Section.
3) Add a section of conclusions.
Response:
Thanks for your suggestions. We have added the Conclusion Section as followed (Line 435-447):
This study focused on an effective preventive behavior – staying at home – to avoid contracting with COVID-19. Cross-country differences between China and the United States in complying with this behavior has been manifested during peaks in this pandemic. Internal cognitive processes, such as health beliefs and behavioral intention, were suggested to explicate the cross-country differences. The structural equation modeling results showed that the links of health beliefs (i.e., perceived susceptibility, severity, and barriers) and behavioral intention significantly mediated the association between nationality and stay-at-home behaviors. Specifically, American college students perceived high susceptibility of contracting COVID-19 whereas Chinese college students perceived high severity of contracting COVID-19, especially few barriers to staying at home, which further led to increased behavioral intention and frequent actual behaviors. Overall, these findings provide a scientific basis for the explanation mechanism of the cross-country differences in preventive behaviors, which offers practical insights to understanding how to mitigate the global COVID-19 pandemic.
Reviewer 3 Report
The manuscript is of high quality and internal consistency. It begins with a very well-structured introduction, in which it is decided to give importance to the hypotheses and to justify their design.
The meteorological framework, together with the results, is very well structured. The use of structural equation modeling provides a very robust statistical analysis of the information.
The discussion is also very coherent and in line with the theoretical framework. It uses current bibliography and also other bibliography that is necessary to justify the article.
As recommendations, in case they could be useful, I would suggest giving a little more importance to the limitations of the study, creating a subsection for this purpose.
I also consider it necessary to include a final section called "Conclusion", as is usual in this type of article, to provide the reader with a final image of the manuscript and the main results found.
In summary, it is a manuscript of high quality.
I consider that this manuscript should be published.
Author Response
The manuscript is of high quality and internal consistency. It begins with a very well-structured introduction, in which it is decided to give importance to the hypotheses and to justify their design.
The meteorological framework, together with the results, is very well structured. The use of structural equation modeling provides a very robust statistical analysis of the information.
The discussion is also very coherent and in line with the theoretical framework. It uses current bibliography and also other bibliography that is necessary to justify the article.
1) As recommendations, in case they could be useful, I would suggest giving a little more importance to the limitations of the study, creating a subsection for this purpose.
Response:
Thanks a lot for your suggestions. We have added three limitations of this study in the Discussion Section as followed (see Line 390-400):
Several limitations to this study should be noted. First, bias might occur when measures were used in different cultures (He & van de Vijver, 2012) although additional tests on the measure equivalence showed strong invariance. Future research could conduct pilot studies before formal surveys to further enhance the reliability and validity of measures when used in different cultures. Second, this study recruited only college students from one typical collectivist country and one typical individualistic country; thus, the generalization to other populations and other countries should be made with caution. Future studies could investigate more participants in more countries oriented in corresponding cultures. Third, this study focused only on a specific preventive behavior (i.e., staying at home), which might provide limited contributions to halting the COVID-19 pandemic. Future research could consider other effective preventive measures (e.g., wearing facial masks) for a comprehensive understanding of overcoming this global challenge.
References:
He, J., & van de Vijver, F. (2012). Bias and equivalence in cross-cultural research. Online Readings in Psychology and Culture, 2(2). https://doi.org/10.9707/2307-0919.1111
2) I also consider it necessary to include a final section called "Conclusion", as is usual in this type of article, to provide the reader with a final image of the manuscript and the main results found.
Response:
Thanks for your suggestions. We have added the Conclusion Section as followed (Line 435-447):
This study focused on an effective preventive behavior – staying at home – to avoid contracting with COVID-19. Cross-country differences between China and the United States in complying with this behavior has been manifested during peaks in this pandemic. Internal cognitive processes, such as health beliefs and behavioral intention, were suggested to explicate the cross-country differences. The structural equation modeling results showed that the links of health beliefs (i.e., perceived susceptibility, severity, and barriers) and behavioral intention significantly mediated the association between nationality and stay-at-home behaviors. Specifically, American college students perceived high susceptibility of contracting COVID-19 whereas Chinese college students perceived high severity of contracting COVID-19, especially few barriers to staying at home, which further led to increased behavioral intention and frequent actual behaviors. Overall, these findings provide a scientific basis for the explanation mechanism of the cross-country differences in preventive behaviors, which offers practical insights to understanding how to mitigate the global COVID-19 pandemic.
In summary, it is a manuscript of high quality.
I consider that this manuscript should be published.